# Meta-Consolidation for Continual Learning

**K J Joseph and Vineeth N Balasubramanian**

Department of Computer Science and Engineering
Indian Institute of Technology Hyderabad, India
`{cs17m18p100001,vineethnb}@iith.ac.in`

## Abstract

The ability to continuously learn and adapt itself to new tasks, without losing grasp of already acquired knowledge is a hallmark of biological learning systems, which current deep learning systems fall short of. In this work, we present a novel methodology for continual learning called MERLIN: Meta-Consolidation for Continual Learning. We assume that weights of a neural network $\psi$, for solving task $t$, come from a meta-distribution $p(\psi|t)$. This meta-distribution is learned and consolidated incrementally. We operate in the challenging online continual learning setting, where a data point is seen by the model only once. Our experiments with continual learning benchmarks of MNIST, CIFAR-10, CIFAR-100 and Mini-ImageNet datasets show consistent improvement over five baselines, including a recent state-of-the-art, corroborating the promise of MERLIN.

## 1  Introduction

The human brain is able to constantly, and incrementally, consolidate new information with existing information, allowing for quick recall when needed [5, 78]. In this natural setting, it is not common to see the same data sample multiple times, or even twice at times. Human memory capacity is also limited which forbids memorizing all examples that are seen during its lifetime [39]. Hence, the brain operates in an *online manner*, where it is able to adapt itself to continuously changing data distributions without losing grasp of its previously acquired knowledge [33]. Unfortunately, deep neural networks have been known to suffer from catastrophic forgetting [53, 25], where they fail to retain performance on older tasks, while learning new tasks.

*Continual learning* is a machine learning setting characterized by its requirement to have a learning model incrementally adapt to new tasks, while not losing its performance on previously learned tasks. Note that 'task' here can refer to a set of new classes, new domains (e.g. thermal, RGB) or even new tasks in general (e.g. colorization, segmentation) [63, 68, 89]. The last few years have seen many efforts to develop methods to address this setting from various perspectives. One line of work [88, 38, 47, 2, 16, 71, 56, 13, 79, 42] constrains the parameters of the deep network trained on Task A to not change much while learning a new Task B, while another - replay-based methods - store [63, 48, 14, 66, 9, 3, 4, 22] or generate [74, 81, 46, 45] examples of previous tasks to finetune the final model at evaluation time. Another kind of methods [51, 72, 69, 19, 68, 62, 87] attempt to expand the network to increase the capacity of the model, while learning new tasks. Broadly speaking, all these methods manipulate the data space or the weight space in different ways to achieve their objectives.

In this work, we propose a different perspective to addressing continual learning, based on the latent space of a weight-generating process, rather than the weights themselves. Studies of the human brain suggest that knowledge and skills to solve tasks are represented in a meta-space of concepts with a high-level semantic basis [28, 10, 50]. The codification from tasks to concepts, and the periodic consolidation of memory, are considered essential for a transferable and compact representation of knowledge that helps humans continually learn [11, 85, 5]. Current continual learning methods consolidate (assimilate knowledge on past tasks) either in the weight space [13,

38, 47, 16, 88, 56, 2, 71] or in the data space [14, 63, 74, 48, 3, 4, 45, 66]. Even meta-learning based continual learning methods that have been proposed in the recent past [34, 24, 6, 64], meta-learn an initialization amenable for quick adaptation across tasks, similar to MAML [23], and hence operate in the weight space. We propose MERLIN: Meta-Consolidation for Continual Learning, a new method for continual learning that is based on consolidation in a meta-space, viz. the latent space which generates model weights for solving downstream tasks.

We consider weights of a neural network $\psi$, which can solve a specific task, to come from a meta-distribution $p(\psi|t)$, where $t$ is a representation for the task. We propose a methodology to learn this distribution, as well as continually adapt it to be competent on new tasks by consolidating this meta-space of model parameters whenever a new task arrives. We refer to this process as "Meta-Consolidation". We find that *continually learning in the parameter meta-space with consolidation* is an effective approach to continual learning. Learning such a meta-distribution $p(\psi|t)$ provides additional benefits: (i) at inference time, any number of models can be sampled from the distribution $\psi_t \sim p(\psi|t)$, which can then be ensembled for prediction in each task (Sec 4.1.5); (ii) it is easily adapted to work in multiple settings such as class-incremental and domain-incremental continual learning (Sec 4); and (iii) it can work in both a task-aware setting (where the task is known at test time) and task-agnostic setting where the task is not known at test time (achieved by marginalizing over $t$, Sec 5.2). Being able to take multiple passes through an entire dataset is an assumption that most existing continual learning methods make [2, 38, 47, 88, 63, 12]. Following [4, 3, 14, 48], we instead consider the more challenging (and more natural) online continual learning setting where 'only a single pass through the data' is allowed. We compare MERLIN against a recent state-of-the-art GSS [4], as well as well-known methods including GEM [48], iCaRL [63] and EWC [38] on Split MNIST [13], Permuted MNIST [88], Split CIFAR-10 [88], Split CIFAR-100 [63] and Split Mini-Imagenet [15] datasets. We observe consistent improvement across the datasets over baseline methods in Sec 4.

The key contributions of this work can be summarized as: (i) We introduce a new perspective to continual learning based on the meta-distribution of model parameters, and their consolidation over tasks arriving in time; (ii) We propose a methodology to learn this distribution using a Variational Auto-encoder (VAE) [37] with learned task-specific priors, which also allows us to ensemble models for each task at inference; (iii) We show that the proposed method outperforms well-known benchmark methods [48, 63, 38], as well as a recent state-of-the-art method [4], on five continual learning datasets; (iv) We perform comprehensive ablation studies to provide a deeper understanding of the proposed methodology and showcase its usefulness. To the best of our knowledge, MERLIN is the first effort to incrementally learn in the meta-space of model parameters.

## 2   Related Work

In this section, we review existing literature that relate to our proposed methodology from two perspectives: continual learning and meta-learning. We also discuss connections to neuroscience literature in the Appendix.

**Continual learning methods:** In continual learning, when a new task $t_k$ comes by, the weights of the associated deep neural network, $\psi_{t_k}$, get adapted for the new task causing the network to perform badly on previously learned tasks. To overcome this issue (called catastrophic forgetting [53, 25]), one family of existing efforts [2, 38, 47, 88] force the newly learned weight configuration $\psi_{t_k}$, to be close to the previous weight configuration $\psi_{t_{k-1}}$, so that the performance on both tasks are acceptable. This approach, by design, can restrict the freedom of the model to learn new tasks. Another set of methods [14, 63, 48, 12] store a few samples (called exemplars) in a fixed-size, task-specific episodic memory, and use strategies like distillation [31] and finetuning to ensure that $\psi_{t_k}$ performs well on all tasks seen so far. These methods work at the risk of memorization. Shin *et al.* [74] instead used a generative model to synthesize examples for all the previous tasks, which allows generating infinite data for the seen tasks. However, as more tasks are added, the capacity of the model reduces, and the generative model does not work well in practice. A recent group of methods [51, 72, 69] attempt to expand the network dynamically to accommodate learning new tasks. While this is an interesting approach, the model size in such methods can increase significantly, hampering scalability.

The aforementioned methods operate in an offline fashion, wherein once data for a specific task is available, it can be iterated over multiple times to learn the task. In contrast, online continual learning methods [4, 3, 14, 48] tackle a more challenging setting, closer to how humans operate, where all datapoints are seen only once during the lifetime of the model. [4, 3] proposes methods to choose

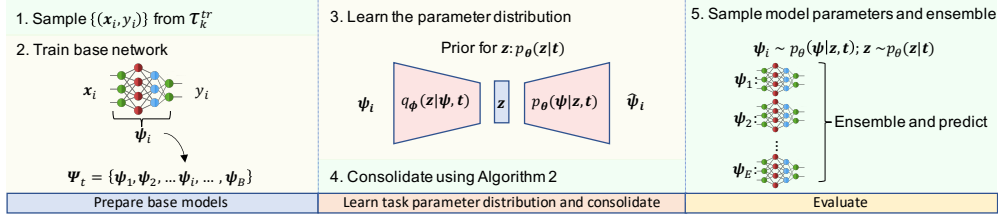

Figure 1: Overview of MERLIN. While learning a new task $\boldsymbol{\tau}_k$ at time $k$, $B$ models are trained and saved to $\boldsymbol{\Psi}_t$. These weights are used to learn a task-specific parameter distribution with task-specific priors in Step 3. In Step 4, the model is consolidated by replaying parameters generated using earlier task-specific priors as in Algorithm 2. At test time, model parameters are sampled from $p_{\boldsymbol{\theta}}(\boldsymbol{\phi}|\boldsymbol{z}, \boldsymbol{t})$ and ensembled.

optimal training datapoints for learning continually, which they found is critical in an online continual learning setting, while [14, 48] constrains the learning of the new task, such that the loss on the previously learned class should not increase. We refer the interested reader to [17, 59] for a detailed survey of current continual learning methods and practices. Contrary to these methods, we propose a new perspective to continual learning based on meta-consolidation in the parameter space which has its own benefits as mentioned earlier.

**Meta-learning methods:** Meta-learning encapsulates a wide variety of methods that can adapt to new tasks with very few training examples. Meta-learning algorithms - which mostly have focused on the standard learning setting (not continual learning) - can be broadly classified into black-box adaptation based methods [70, 55], optimization based [23, 57, 90] and non-parametric methods [83, 75, 77]. Hypernetworks [27] is one black-box adaptation method where a meta-neural network is used to predict the weights of another neural network. This opened up an interesting research direction, in exploiting the meta-manifold of model parameters for multi-task learning [67], neural architecture search [8], zero-shot learning [58, 82] and style transfer [73] in recent years. One can view our approach as having a similar view to online continual learning (which has not been done before), although our methodology of learning the distribution of weights of the network is different from [27] and allows more flexibility in sampling and generating multiple models at inference. Mandivarapu *et al.* [52] auto-encode the weights in a continual setting, but the size of their meta-model is many times the size of the task network and does-not operate in an online setting. The work closest to ours is by Johannes *et al.* [84], which recently tried using HyperNetworks in the context of continual learning. We differ from [84] in the following ways: (i) we operate in the more natural and challenging *online* continual learning setting, while they use multiple passes over the data (ii) they learn a deterministic function which when conditioned on a task embedding, generates target weights, while we model the weight distribution itself, which allows us to sample as many model parameters at inference time and ensemble them.

## 3 MERLIN: Methodology

We begin by summarizing the overall methodology of MERLIN, as in Algorithm 1 and Fig 1. We consider a sequence of tasks $\boldsymbol{\tau}_1, \boldsymbol{\tau}_2, \cdots \boldsymbol{\tau}_{k-1}$ that have been seen by the learner, until now. A new task $\boldsymbol{\tau}_k$ is introduced at time instance $k$. (Note that task can refer to a new class or a domain, both of which are studied in Sec 4.) Each task, $\boldsymbol{\tau}_j, j \in \{1, \cdots, k\}$ consists of $\boldsymbol{\tau}_j^{tr}, \boldsymbol{\tau}_j^{val}$ and $\boldsymbol{\tau}_j^{test}$ corresponding to training, validation and test samples for that task. Each task $\boldsymbol{\tau}_j$ is also represented by a corresponding vector $\boldsymbol{t}_j$, as described in Sec 3.1. As the first step, we train a set of $B$ base models on random subsets of $\boldsymbol{\tau}_k^{tr}$ (sampled without replacement, note that this does not violate the online setting as this is done by choosing each streaming point with a certain probability in each base model) to obtain a collection of models $\boldsymbol{\Psi}_k = \{\psi_k^1, \cdots, \psi_k^B\}$ (as in first section of Fig 1, line 2 in Algorithm 1). $\boldsymbol{\Psi}_k$ is then used to learn a task-specific parameter distribution $p_{\boldsymbol{\theta}}(\boldsymbol{\psi}_k|\boldsymbol{t}_k)$ using a VAE-like strategy further described in Sec 3.1. This is followed by a meta-consolidation phase, where we sample model parameters for all tasks seen so far, from the decoder of the VAE, each conditioned on a task-specific prior, and use them to refine the overall VAE, as described in Sec 3.2. At inference, we sample models from the parameter distributions for each task, and use them to evaluate on the test data following the methodology described in Sec 3.3. A set of $\psi$s are sampled from $p_{\boldsymbol{\theta}}(\boldsymbol{\psi}|\boldsymbol{z}, \boldsymbol{t})$ for task $\boldsymbol{t}$ and ensembled to obtain the final result. We now describe each of the components below.

**3.1 Modeling Task-specific Parameter Distributions:** Each task $\boldsymbol{\tau}_j$ is represented by a corresponding vector representation $\boldsymbol{t}_j$. Our framework allows any fixed-length vector representation

---

**Algorithm 1** MERLIN: Overall Methodology

---

**Input:** Tasks: $\mathcal{T} = \{\boldsymbol{\tau}_1 \cdots \boldsymbol{\tau}_k\}$; Task training data: $\boldsymbol{\tau}_j^{tr} = \{(\boldsymbol{x}_i, y_i)\}_{i=1}^{N_{t_j}^j}$; # of base models: $B$

1: **for** j = 1 to $|\mathcal{T}|$ **do**
2:     $\boldsymbol{\Psi}_j \leftarrow$ Train $B$ models by randomly sub-sampling training data from $\boldsymbol{\tau}_j^{tr}$.
3:     Learn task parameter distribution and task specific prior using methodology in Sec 3.1 and $\boldsymbol{\Psi}_j$.
4:     Consolidate using methodology in Sec 3.2, Algorithm 2.
5: Perform inference on $\mathcal{T}$, using methodology in Sec 3.3, Algorithm 3.

---

for this task descriptor, including semantic embeddings such as GloVe [61] or Word2Vec [54], or simply a one-hot encoding of the task. A richer embedding such as Task2Vec [1], which takes into account task correlations, may also be used. Details of our implementation are deferred to Sec 4. We consider $\boldsymbol{\psi_t}$, the weights of a neural network which can solve a task $\boldsymbol{t}$, as generated by a random process given by a continuous latent variable $\boldsymbol{z}$, i.e. $\boldsymbol{\psi_t}$ is generated from $p_{\boldsymbol{\theta}^*}(\boldsymbol{\psi}|\boldsymbol{z}, \boldsymbol{t})$, where $\boldsymbol{z}$ is sampled from a conditional prior distribution $p_{\boldsymbol{\theta}^*}(\boldsymbol{z}|\boldsymbol{t})$ (We denote the $j^{\text{th}}$ task, $\boldsymbol{t}_j$ as $\boldsymbol{t}$ for simplicity of explaining the model in this subsection). $\boldsymbol{\theta}^*$ refers to the unknown optimal parameters of the weight-generating distribution, which we seek to find. We achieve this objective using a VAE-like formulation [37], adapted for this problem. Computing the marginal likelihood of the weight distribution $p_{\boldsymbol{\theta}}(\boldsymbol{\psi}|\boldsymbol{t}) = \int p_{\boldsymbol{\theta}}(\boldsymbol{\psi}|\boldsymbol{z}, \boldsymbol{t}) p_{\boldsymbol{\theta}}(\boldsymbol{z}|\boldsymbol{t}) d\boldsymbol{z}$ is intractable as its true posterior $p_{\boldsymbol{\theta}}(\boldsymbol{z}|\boldsymbol{\psi}, \boldsymbol{t}) = \frac{p_{\boldsymbol{\theta}}(\boldsymbol{\psi}|\boldsymbol{z},\boldsymbol{t}) p_{\boldsymbol{\theta}}(\boldsymbol{z}|\boldsymbol{t})}{p_{\boldsymbol{\theta}}(\boldsymbol{\psi}|\boldsymbol{t})}$ is intractable to compute. We introduce the approximate posterior $q_{\boldsymbol{\phi}}(\boldsymbol{z}|\boldsymbol{\psi}, \boldsymbol{t})$, parametrized by $\boldsymbol{\phi}$, as a variational distribution for the intractable true posterior $p_{\boldsymbol{\theta}}(\boldsymbol{z}|\boldsymbol{\psi}, \boldsymbol{t})$. The marginal likelihood of the parameter distribution, $\log p_{\boldsymbol{\theta}}(\boldsymbol{\psi}|\boldsymbol{t})$, can then be written as (please refer the Appendix for the complete derivation):

$$\log p_{\boldsymbol{\theta}}(\boldsymbol{\psi}|\boldsymbol{t}) = D_{KL}(q_{\boldsymbol{\phi}}(\boldsymbol{z}|\boldsymbol{\psi}, \boldsymbol{t}) \, || \, p_{\boldsymbol{\theta}}(\boldsymbol{z}|\boldsymbol{\psi}, \boldsymbol{t})) + \underbrace{\int_{\boldsymbol{z}} q_{\boldsymbol{\phi}}(\boldsymbol{z}|\boldsymbol{\psi}, \boldsymbol{t}) \, \log \frac{p_{\boldsymbol{\theta}}(\boldsymbol{z}, \boldsymbol{\psi}|\boldsymbol{t})}{q_{\boldsymbol{\phi}}(\boldsymbol{z}|\boldsymbol{\psi}, \boldsymbol{t})}}_{\mathcal{L}(\boldsymbol{\theta}, \boldsymbol{\phi}|\boldsymbol{\psi}, \boldsymbol{t})} \tag{1}$$

Similar to a VAE, we can maximize the log likelihood by maximizing the lower bound. $\mathcal{L}(\boldsymbol{\theta}, \boldsymbol{\phi}|\boldsymbol{\psi}, \boldsymbol{t})$ can hence be rewritten as (complete derivation in the Appendix):

$$\mathcal{L}(\boldsymbol{\theta}, \boldsymbol{\phi}|\boldsymbol{\psi}, \boldsymbol{t}) = -D_{KL}(q_{\boldsymbol{\phi}}(\boldsymbol{z}|\boldsymbol{\psi}, \boldsymbol{t}) \, || \, p_{\boldsymbol{\theta}}(\boldsymbol{z}|\boldsymbol{t})) + \mathbb{E}_{q_{\boldsymbol{\phi}}(\boldsymbol{z}|\boldsymbol{\psi}, \boldsymbol{t})}[\log p_{\boldsymbol{\theta}}(\boldsymbol{\psi}|\boldsymbol{z}, \boldsymbol{t})] \tag{2}$$

where the second term is the expected negative reconstruction error. The KL divergence term on the RHS forces the approximate posterior of weights to be close to a *task-specific prior* $p_{\boldsymbol{\theta}}(\boldsymbol{z}|\boldsymbol{t})$. Note that this differentiates this formulation from a conditional VAE [36], where the latents are directly conditioned. In our model, the latents are conditioned indirectly through priors that are conditioned on task vectors. Assuming $q_{\boldsymbol{\phi}}(.)$ and $p_{\boldsymbol{\theta}}(.)$ to be Gaussian distributions, the KL divergence term can be computed in closed form. The second term requires sampling, and the model parameters, $\boldsymbol{\phi}$ and $\boldsymbol{\theta}$, are trained using the reparameterization trick [37], backpropagation and Stochastic Gradient Descent.

Instead of choosing the prior to be an isotropic multivariate Gaussian $p_{\boldsymbol{\theta}}(\boldsymbol{z}) = \mathcal{N}(\boldsymbol{z}|\boldsymbol{0}, \mathbf{I})$ as in a vanilla VAE, we use a learned task-specific prior, $p_{\boldsymbol{\theta}}(\boldsymbol{z}|\boldsymbol{t})$, given by:

$$p_{\boldsymbol{\theta}}(\boldsymbol{z}|\boldsymbol{t}) = \mathcal{N}(\boldsymbol{z}|\boldsymbol{\mu_t}, \boldsymbol{\Sigma_t}); \text{ where } \boldsymbol{\mu_t} = \boldsymbol{W}_\mu^T \boldsymbol{t} \text{ and } \boldsymbol{\Sigma_t} = \boldsymbol{W}_\Sigma^T \boldsymbol{t} \tag{3}$$

Here, $\boldsymbol{W}_\mu$ and $\boldsymbol{W}_\Sigma$ are parameters which are learned alongside $\boldsymbol{\phi}$ and $\boldsymbol{\theta}$, when maximizing the lower bound (Eqn 2) using backpropagation. We find that a simple linear model is able to learn $\boldsymbol{W}_\mu$ and $\boldsymbol{W}_\Sigma$ effectively, as revealed in our experiments.

As already stated, the probabilistic encoder $q_{\boldsymbol{\phi}}(\boldsymbol{z}|\boldsymbol{\psi}, \boldsymbol{t})$ and decoder $p_{\boldsymbol{\theta}}(\boldsymbol{\psi}|\boldsymbol{z}, \boldsymbol{t})$ are materialized using two neural networks parametrized by $\boldsymbol{\phi}$ and $\boldsymbol{\theta}$ respectively. To train these, we consider $\boldsymbol{\Psi}_j = \{\boldsymbol{\psi}_j^i\}_{i=1}^B$, a set of $B$ model parameters that are obtained by training on the $j^{\text{th}}$ task by sampling the corresponding training data without replacement. These models are now used to learn the "meta-parameters" $\boldsymbol{\phi}$ and $\boldsymbol{\theta}$, as well as $\boldsymbol{\mu}_j$ and $\boldsymbol{\Sigma}_j$ of the Gaussian prior, specific to the task. At inference, one can sample as many $\psi$s (task-specific models) as needed from the decoder using different samples from the task-specific prior. Obtaining a model $\boldsymbol{\psi}$ for each task $\boldsymbol{t}$ from the parameter distribution $p_{\boldsymbol{\theta}}(\boldsymbol{\psi}|\boldsymbol{z}, \boldsymbol{t})$, is a two-step process: (i) Sample $\boldsymbol{z}$ from task-specific prior distribution i.e. $\boldsymbol{z} \sim \mathcal{N}(\boldsymbol{z}|\boldsymbol{\mu_t}, \boldsymbol{\Sigma_t})$; (ii) Sample $\boldsymbol{\psi}$ from the probabilistic decoder, using $\boldsymbol{z}$ i.e. $\boldsymbol{\psi} \sim p_{\boldsymbol{\theta}}(\boldsymbol{\psi}|\boldsymbol{z}, \boldsymbol{t})$. Note that we have only a single VAE model, which can generate network parameters for each task individually by conditioning the VAE on the corresponding task-specific prior.

**3.2 Meta-consolidation:** When the continual learner encounters the $k^{\text{th}}$ task and receives the set of optimal model parameters for this task $\boldsymbol{\Psi}_k = \{\boldsymbol{\psi}_k^i\}_{i=1}^B$, directly updating the VAE on $\boldsymbol{\Psi}_k$ alone causes a distributional shift towards the $k^{\text{th}}$ task. We address this by "meta-consolidating" the encoder and decoder, after accommodating the new task. Algorithm 2 summarizes the steps involved in this consolidation. We assume that all the learned task-specific priors are stored and available to us. This adds negligible storage complexity as it involves storing only means and covariances. For each task $\boldsymbol{\tau}_1, \cdots, \boldsymbol{\tau}_k$ seen so far, a task-specific $\boldsymbol{z}_t$ is sampled from the corresponding task-specific prior distribution. Next, $P$ pseudo-models are sampled from the decoder $p_{\boldsymbol{\theta}}(\boldsymbol{\psi}|\boldsymbol{z}_t, \boldsymbol{t})$. These generated pseudo-models are used to finetune the parameters of the encoder and the decoder by maximizing the lower bound, $\mathcal{L}(\boldsymbol{\theta}, \boldsymbol{\phi}|\boldsymbol{\psi}, \boldsymbol{t})$ defined in Eqn 2. Note that we update only $\boldsymbol{\theta}$ and $\boldsymbol{\phi}$, and not the parameters of the task-specific priors, which are fixed. One could consider this as a "replay" strategy, only using the meta-parameter space. This ensures that after learning each new task, the encoder and decoder are competent enough to generate model parameters for solving all tasks seen till then. Importantly, we do not learn separate encoder-decoders per task and do not store weight parameters ($\boldsymbol{\psi}_j$) for any task, which makes our proposed method storage-efficient. The consolidation in this step takes place in the task parameter meta-space (unlike earlier methods), which we find to be effective for continual learning, as validated by our experimental results in Sec 4.

---

**Algorithm 2** META-CONSOLIDATION IN MERLIN

---

**Input:** Encoder: $q_{\boldsymbol{\phi}}(\boldsymbol{z}|\boldsymbol{\psi}, \boldsymbol{t})$; Decoder: $p_{\boldsymbol{\theta}}(\boldsymbol{\psi}|\boldsymbol{z}, \boldsymbol{t})$; Last seen task: $\boldsymbol{\tau}_k$; Task priors: $\{\boldsymbol{P}_i\}_{i=1}^k$, $\boldsymbol{P}_i = (\boldsymbol{\mu}_i, \boldsymbol{\Sigma}_i)$;
    # of psuedo-models: $P$
**Output:** Consolidated encoder-decoder parameters: $\boldsymbol{\phi}$ and $\boldsymbol{\theta}$
 1: **for** j = 1 to k **do**
 2:     $\boldsymbol{\mu}_j, \boldsymbol{\Sigma}_j \leftarrow \boldsymbol{P}_j$
 3:     $\boldsymbol{z}_j \sim \mathcal{N}(\boldsymbol{z}|\boldsymbol{\mu}_j, \boldsymbol{\Sigma}_j)$                                           ▷ *Task specific* $\boldsymbol{z}_j$
 4:     $Loss \leftarrow \sum_{i=1}^P \mathcal{L}(\boldsymbol{\theta}, \boldsymbol{\phi}|\boldsymbol{\psi}_i, \boldsymbol{t})$; where $\boldsymbol{\psi}_i \sim p_{\boldsymbol{\theta}}(\boldsymbol{\psi}|\boldsymbol{z}_t, \boldsymbol{t})$       ▷ $\mathcal{L}(.)$ *is defined in Equation 2.*
 5:     $\boldsymbol{g} \leftarrow \nabla_{\boldsymbol{\theta}, \boldsymbol{\phi}} Loss$
 6:     $\boldsymbol{\phi}, \boldsymbol{\theta} \leftarrow$ Update parameters $\boldsymbol{\phi}, \boldsymbol{\theta}$, using gradient $\boldsymbol{g}$.
 7: **return** $\boldsymbol{\phi}, \boldsymbol{\theta}$

---

**3.3 Inference:** Having learned the distribution to generate model parameters for each task seen until now, the learned task-specific priors give us the flexibility to sample any number of models from this distribution at inference/test time. This allows for *ensembling* multiple models at test time (none of which need to be stored a priori), which is a unique characteristic of the proposed method.

Ideally, a continual learning algorithm should be able to solve all tasks encountered so far, without access to any task-specific information at inference time. While a handful of existing methods [29, 63] take this into consideration, many others [48, 38, 88] assume availability of task-identifying information during inference. This is also referred as single-head or multi-head evaluation in literature [13, 80, 21]. Our methodology works with both evaluation settings, which we refer to as *task-agnostic* and *task-aware* inference, as described below. Algorithm 3 outlines the inference steps. We use the

---

**Algorithm 3** MERLIN INFERENCE

---

**Input:** Decoder: $p_{\boldsymbol{\theta}}(\boldsymbol{\psi}|\boldsymbol{z}, \boldsymbol{t})$; Last seen task: $\boldsymbol{\tau}_k$; Task priors: $\boldsymbol{P} = \{\boldsymbol{P}_i\}_{i=1}^k$, $\boldsymbol{P}_i = (\boldsymbol{\mu}_i, \boldsymbol{\Sigma}_i)$; Exemplars:
    $\mathcal{E} = \{Ex_i\}_{i=1}^m$, $Ex_i = \{(\boldsymbol{x}_i, \boldsymbol{y}_i)\}$; Number of base models to ensemble from: $E$
 1: **if** *Task-agnostic inference* **then**                                      ▷ *Task-agnostic inference*
 2:     $\boldsymbol{z} \sim \mathcal{N}(\boldsymbol{z}|\boldsymbol{\mu}, \boldsymbol{\Sigma})$ where $\boldsymbol{\mu} \leftarrow \frac{1}{k}\sum_{i=1}^k \boldsymbol{\mu}_i$ and $\boldsymbol{\Sigma} \leftarrow \frac{1}{k}\sum_{i=1}^k \boldsymbol{\Sigma}_i$
 3:     $\boldsymbol{\Psi} \leftarrow$ Sample $E$ models from $p_{\boldsymbol{\theta}}(\boldsymbol{\psi}|\boldsymbol{z})$
 4:     $\boldsymbol{\Psi} \leftarrow$ Fine-tune $\boldsymbol{\Psi}$ on $\mathcal{E}$
 5:     Ensemble results from $\boldsymbol{\Psi}$ to solve all tasks $(\boldsymbol{\tau}_1, \cdots, \boldsymbol{\tau}_k)$
 6: **if** *Task-aware inference* **then**                                        ▷ *Task-aware inference*
 7:     **for** j = 1 to k **do**
 8:         $\boldsymbol{z}_j \sim \mathcal{N}(\boldsymbol{z}|\boldsymbol{\mu}_j, \boldsymbol{\Sigma}_j)$ where $\boldsymbol{\mu}_j, \boldsymbol{\Sigma}_j \leftarrow \boldsymbol{P}_j$
 9:         $\boldsymbol{\Psi}_j \leftarrow$ Sample $E$ models from $p_{\boldsymbol{\theta}}(\boldsymbol{\psi}|\boldsymbol{z}_j, \boldsymbol{t}_j)$
10:         $\boldsymbol{\Psi}_j \leftarrow$ Fine-tune $\boldsymbol{\Psi}_j$ on $Ex_j$
11:         Ensemble results from $\boldsymbol{\Psi}_j$ to solve task $\boldsymbol{\tau}_j$.

---

consolidated decoder $p_{\boldsymbol{\theta}}(\boldsymbol{\psi}|\boldsymbol{z}, \boldsymbol{t})$, and task-specific priors $\boldsymbol{P}$ at test time. In a *task-agnostic* setting, a single overall prior distribution is aggregated from all task-specific priors. We find that a simple averaging of parameters of the prior distributions works well in practice. The latent variable $\boldsymbol{z}$ is sampled from this aggregated prior. The model parameters sampled from the decoder using $\boldsymbol{z}$ has the

capability to solve all tasks seen till then. The generated $\psi$ is amenable for quick adaptation to each task similar to MAML [23] to obtain task-dependent model parameters. To leverage this, we store a small set of randomly selected exemplars for each task, which is used to finetune $\psi$ for each task. $E$ such models are sampled from the decoder, each of which is finetuned on all tasks. An ensemble of these models is used to predict the final output. *Task-aware* inference works very similarly, with the only change that $z_j$ is sampled from each task-specific prior. The task-specific model parameters $\psi_j$ are then sampled from $p_\theta(\psi|z_j, t_j)$, finetuned on task exemplars $Ex_j$ and finally ensembled. The ensembling step adds minimal inference overhead as explained in Sec 4.1.5.

## 4 Experiments and Results

We evaluate MERLIN against prominent methods for continual learning: GEM [48], iCaRL [63], EWC [38], a recent state-of-the-art GSS [4], and a crude baseline where a single model is trained across tasks (referred to as 'Single' in our results). GSS and GEM are inherently designed for online continual learning, while iCaRL and EWC are easily adapted to this setting following [48, 14]. These baselines consolidate in different spaces: EWC [38], a regularization-based method, consolidates in the weight space; GEM [48], GSS [4], and iCaRL [63] use exemplar memory and consolidate in the data space; while MERLIN consolidates in the meta-space of parameters. We used the official implementations of each of these baseline methods for fair comparison. Five standard continual learning benchmarks, viz. Split MNIST [13], Permuted MNIST [88], Split CIFAR-10 [88], Split CIFAR-100 [63] and Split Mini-Imagenet [15], are used in the experiments, following recent continual learning literature [12, 4, 63, 51, 13].

**4.1 Experimental Setup:** We describe the datasets, evaluation metrics and other implementation details below, before presenting our results. Our code[1] is implemented in PyTorch [60] and runs on a single NVIDIA V-100 GPU.

**4.1.1 Datasets:** We describe the datasets considered briefly below (a detailed description is presented in the Appendix):

*Split MNIST and Permuted MNIST:* Subsequent classes in the MNIST [43] dataset are paired together and presented as a task in *Split MNIST*, a well-known benchmark for continual learning. This results in 5 incremental tasks. In *Permuted MNIST*, each task is a unique spatial permutation of $28 \times 28$ images of MNIST. 10 such permutations are generated to create 10 tasks. We use 1000 images per task for training and the model is evaluated on the all test examples, following the protocol in [48].

*Split CIFAR-10 and Split CIFAR-100:* 5 and 10 tasks are created by grouping together 2 and 10 classes from CIFAR-10 [41] and CIFAR-100 [41] datasets respectively. Following [48], 2500 examples are used per task for training. Trained models are evaluated on the whole test set.

*Split Mini-Imagenet:* Mini-Imagenet [83] is a subset of ImageNet [18] with a total of 100 classes and 600 images per class. Each task consists of 10 disjoint subset of classes from these 100 classes. Similar to CIFAR variants, 2500 examples per task is used for training, as in [15].

We note a distinction between tasks in "Split" versions of the datasets and Permuted MNIST. In "Split" datasets, the label space expands with tasks, while for Permuted MNIST, the data space changes with tasks without changing the label space. The former is referred to as the *Class-Incremental* setting, while the latter as *Domain-Incremental* setting in literature [32, 80]. MERLIN works on both settings.

**4.1.2 Base Classifier Architecture:** For CIFAR and Mini-ImageNet datasets, a modified ResNet [30] architecture is used, which is 10 layers deep and has fewer number of feature maps in each of the four residual blocks $(5, 10, 20, 40)$. This reduces the number of parameters from $0.27M$ to $34997$. In spite of using a weaker base network (owing to computing constraints), our method outperforms baselines, as shown in our results. For the MNIST dataset, we use a two-layer fully connected neural network with 100 neurons each, with ReLU activation, following the experimental setting in GEM [48]. To train these base models (which are then used to train the VAE in MERLIN), batch size is set to 10 and Adam [35] is used as the optimizer, with an initial learning rate of 0.001 and weight decay of 0.001. To ensure the online setting, the model is trained only for a single epoch, similar to baseline methods [48, 4, 3]. For class-incremental experiments, we follow earlier methods [48, 4] to assume an upper-bound on the number of classes to expect, and modify the loss function to consider only classes that are seen so far. This is done by setting the final layer classification logits of the unseen class to a very low value $(-10^{10})$, as in [48, 14, 62].

| Datasets → | Split MNIST | | Permuted MNIST | | Split CIFAR-10 | | Split CIFAR-100 | | Split Mini-ImageNet | |
|---|---|---|---|---|---|---|---|---|---|---|
| Methods ↓ | A ($\uparrow$) | F ($\downarrow$) | A ($\uparrow$) | F ($\downarrow$) | A ($\uparrow$) | F ($\downarrow$) | A ($\uparrow$) | F ($\downarrow$) | A ($\uparrow$) | F ($\downarrow$) |
| Single | $44.8 \pm 0.3$ | $98.3 \pm 0.5$ | $73.1 \pm 2.2$ | $15.7 \pm 1.9$ | $73.2 \pm 3.1$ | $12.6 \pm 4.4$ | $30.8 \pm 3.5$ | $20.5 \pm 2.6$ | $27.5 \pm 2.6$ | $17.1 \pm 2.7$ |
| EWC [38] | $45.1 \pm 0.1$ | $98.4 \pm 0.2$ | $74.9 \pm 2.1$ | $12.4 \pm 2.5$ | $74.2 \pm 2.2$ | $14.5 \pm 3.4$ | $29.2 \pm 3.3$ | $22.5 \pm 4.1$ | $28.1 \pm 2.5$ | $18.0 \pm 4.6$ |
| GEM [48] | $86.7 \pm 1.5$ | $23.4 \pm 1.8$ | $82.5 \pm 4.9$ | $0.8 \pm 0.4$ | $79.1 \pm 1.6$ | $5.9 \pm 1.7$ | $40.6 \pm 1.9$ | $1.3 \pm 1.8$ | $34.1 \pm 1.2$ | $4.7 \pm 0.9$ |
| iCaRL [63] | $89.9 \pm 0.9$ | $\mathbf{1.7 \pm 1.3}$ | - | - | $72.6 \pm 1.3$ | $4.1 \pm 1.5$ | $27.1 \pm 2.9$ | $\mathbf{1.2 \pm 1.3}$ | $38.8 \pm 1.6$ | $3.5 \pm 0.6$ |
| GSS [4] | $88.3 \pm 0.8$ | $33.3 \pm 2.4$ | $81.4 \pm 1.2$ | $8.8 \pm 1.1$ | $57.9 \pm 2.6$ | $49.2 \pm 7.6$ | $19.1 \pm 0.7$ | $42.7 \pm 1.5$ | $14.8 \pm 0.9$ | $31.3 \pm 3.2$ |
| MERLIN | $\mathbf{90.7 \pm 0.8}$ | $6.4 \pm 1.2$ | $\mathbf{85.5 \pm 0.5}$ | $\mathbf{0.4 \pm 0.4}$ | $\mathbf{82.9 \pm 1.2}$ | $\mathbf{-0.9 \pm 1.9}$ | $\mathbf{43.5 \pm 0.6}$ | $2.9 \pm 3.7$ | $\mathbf{40.1 \pm 0.9}$ | $\mathbf{2.8 \pm 3.2}$ |

Table 1: Average accuracy (A) and average forgetting measure (F) of five baseline methods and MERLIN, across five datasets. MERLIN consistently outperforms the baselines across datasets.

**4.1.3 Task Descriptors:** The inputs to train models for each task are training data points $\boldsymbol{\tau}_k^{tr} = \{(\boldsymbol{x}_i, y_i)\}_{i=1}^{N_{tr}^k}$, corresponding to task $\boldsymbol{\tau}_k$. As in Sec 3.1, in order to condition the prior distribution $p_{\boldsymbol{\theta}}(\boldsymbol{z}|\boldsymbol{t})$ on a task, each task, $\boldsymbol{\tau}_k$, is represented by a corresponding fixed-length task descriptor $\boldsymbol{t}_k \in \mathbb{R}^D$. We use the simplest approach in our experiments: one-hot encoding of the task sequence number. Using semantic task descriptors such as GloVe or Word2Vec can allow our framework to be extended to few-shot/zero-shot continual learning. The label embedding of a zero/few-shot task can be used to condition the prior, which can subsequently generate help decoder models for the zero/few-shot task. We leave this as a direction of future work at this time.

**4.1.4 Training the VAE:** The first step of MERLIN is to train task-specific classification models from training data $\boldsymbol{\tau}_j^{tr}$ using the abovementioned architectures. The weights of these models are used to train the VAE. 10 models are learned for each task by random sampling of subsets (with replacement) from $\boldsymbol{\tau}_k^{tr}$. Considering that base classification models can be large, in order to not make the VAE too large, we use a chunking trick proposed by Johannes *et al.* in [84]. The weights of the base classification models are flattened into a single vector and split into equal sized chunks (last chunk zero-padded appropriately). We use a chunk size of 300 for all experiments, and show a sensitivity analysis on the chunk size in Sec 5.6. The VAE is trained on the chunks (instead of the full models) for scalability, conditioned additionally on the chunk index. At inference, the classifier weights are assembled back by concatenating the chunks generated by the decoder, conditioned on the chunk index. We observed that this strategy worked rather seamlessly, as shown in our results. The approximate posterior $q_{\boldsymbol{\phi}}(\boldsymbol{z}|\boldsymbol{\psi}, \boldsymbol{t})$, is assumed to be a 2-D isotropic Gaussian, whose parameters are predicted using a encoder network with 1 fully connected layer of 50 neurons, followed by two layers each predicting the mean and the covariance vectors. The decoder $p_{\boldsymbol{\theta}}(\boldsymbol{\psi}|\boldsymbol{z}, \boldsymbol{t})$ mirrors the encoder's architecture. The network that generates the mean and diagonal covariance vectors of the learned prior, as in Eqn 3, is modeled as a linear network. AdaGrad [20] is used as the optimizer with an initial learning rate of 0.001. Batch size is set to 1 and the VAE network is trained for 25 epochs.

**4.1.5 Inference:** At test time, we sample 30 models from the trained decoder $p_{\boldsymbol{\theta}}(\boldsymbol{\psi}|\boldsymbol{z}, \boldsymbol{t})$ to solve each task. Algorithm 3 shows how these are obtained for task-aware and task-agnostic settings. These models are ensembled using majority voting. An ablation study varying the number of sampled models at inference is presented in Sec 5.4. We ensure that this adds minimal inference overhead by loading sampled weights into the model sequentially and saving only the final logits for ensembling. Hence, only one model is stored at a given time in memory, allowing our framework to scale up effectively. Any additional steps in our methodology is offset by the choice of very small models to train, and our chunking strategy, leading to minimal overhead in training time over the baseline methods. At test time, our method is real-time, and has no difference from baseline methods.

**4.1.6 Evaluation Metrics:** For all experiments, we consider the *average accuracy* across tasks and *average forgetting measure* as the evaluation criteria, following previous works [13, 48]. Average accuracy ($A \in [0, 100]$) after learning the $k^{th}$ task ($\boldsymbol{\tau}_k$), is defined as $A_k = \frac{1}{k} \sum_{j=1}^{k} a_{k,j}$; where $a_{k,j}$ is the performance of the model on the test set of task $j$, after the model is trained on task $k$. Forgetting is defined as the difference in performance from the peak accuracy of a model learned exclusively for a task and the accuracy of the continual learner on the same task. Average forgetting measure ($F \in [-100, 100]$) after learning the $k^{th}$ task can be defined as $F_k = \frac{1}{k-1} \sum_{j=1}^{k-1} \left( \max_{l \in \{1, \cdots, k-1\}} a_{l,j} - a_{k,j} \right)$.

We ran each experiment with five different seed values and report the mean and standard deviation.

**4.2 Results:** Table 1 shows how MERLIN compares against baseline methods on all the considered datasets. iCaRL [63] results are not reported for Permuted MNIST as it is not a Domain-Incremental methodology. We see that the average accuracy (higher is better) obtained by MERLIN shows significant improvement over baseline methods across datasets and settings. The difference is larger

with increasing complexity of the dataset. MERLIN also shows strong improvement in the forgetting metric on most datasets, including a negative value (showing improved performance on earlier tasks, known as positive backward transfer [14]) on Split CIFAR10. While learning a new task, iCaRL uses distillation loss [31] to enforce that logits of exemplars from the previous tasks, do not alter much with the current task learning. This, we believe, helps iCaRL achieve lower forgetting, when compared to other methods. MERLIN is able to outperform iCaRL too, in two out of four datasets, with reduced forgetting. GSS [4] failed on CIFAR-100 and Mini-ImageNet, even after we tried different hyperparameters. The original paper does not report results on these datasets. We did not include comparison with A-GEM [14] as it reports only marginal improvement over GEM [48]. The evolution of test accuracy with addition of tasks, which showed improvement in accuracy over the baselines from the very first task, is presented in the Appendix due to space constraints.

## 5 Discussions and Analysis

We analyze the effectiveness of MERLIN using more studies, as described below.

**5.1 Is the Probabilistic Decoder Learning the Parameter Distribution?** The key component of MERLIN is the decoder that models the parameter distribution $p_\theta(\psi|z_t, t)$. Should the decoder fail to learn, then each model $\psi_i \sim p_\theta(\psi|z_t, t)$ would be an ineffective sample from an untrained decoder. In such a scenario, the only reason why inference (Algo 3) would work would be due to finetuning of these sampled models on the exemplars (L4, L10; Algo 3). In order to study this, we run an experiment where we intentionally skip training the VAE and the consolidation step that follows. With all other components the same, this corresponds to only finetuning with exemplars. The 'w/o training VAE' row in Tab 2 shows the results. Consistently, on all datasets, the performance drops significantly while removing this key component. This suggests that the proposed meta-consolidation through the VAE is actually responsible for the performance in Tab 1.

**5.2 Task Agnostic vs Task Aware Inference:** All results of MERLIN in Tab 1 do not assume task information during evaluation, and hence operate in the task-agnostic setting. MERLIN can work in both task-agnostic and task-aware settings, as shown in the last two rows of Tab 2. Expectedly, access to task information during inference boosts performance for simple datasets like Split MNIST. For other datasets, average accuracy and forgetting measure is almost similar for both. This supports our choice of aggregation strategy of priors across tasks, which captures the task-agnostic setting quite well.

| Datasets → | Split MNIST | | Permuted MNIST | | Split CIFAR-10 | | Split CIFAR-100 | | Split Mini-ImageNet | |
|---|---|---|---|---|---|---|---|---|---|---|
| Methods ↓ | A (↑) | F (↓) | A (↑) | F (↓) | A (↑) | F (↓) | A (↑) | F (↓) | A (↑) | F (↓) |
| w/o training VAE | 44.8 ± 5.7 | 19.9 ± 2.3 | 29.9 ± 4.7 | 5.8 ± 0.2 | 53.5 ± 6.6 | 3.8 ± 1.3 | 15.7 ± 1.63 | 3.8 ± 0.6 | 16.1 ± 1.9 | 3.4 ± 3.1 |
| Task Agnostic | 90.7 ± 0.8 | 6.4 ± 1.2 | 85.5 ± 0.5 | 0.4 ± 0.4 | 82.9 ± 1.2 | -0.9 ± 1.9 | 43.5 ± 0.6 | 2.9 ± 3.7 | 40.1 ± 0.9 | 2.8 ± 3.2 |
| Task Aware | 97.4 ± 0.3 | 0.1 ± 0.3 | 85.1 ± 0.4 | 1.1 ± 0.8 | 82.3 ± 1.1 | -0.4 ± 1.2 | 44.4 ± 2.8 | 1.7 ± 0.7 | 41.8 ± 1.5 | 1.1 ± 0.8 |

Table 2: MERLIN variants: (i) without training VAE and subsequent consolidation (ii) Task-agnostic (iii) Task-aware MERLIN. Task-agnostic MERLIN performs almost well as Task-aware MERLIN on several datasets.

**5.3 Visualization of Task-specific Priors and Sampled Models:** In Fig 2, we visualize the latent variables sampled from the learned prior distribution $z_t \sim p_\theta(z|t)$ *(left)* - which are two-dimensional themselves, and the 2D t-SNE embeddings [49] of their corresponding model parameters generated using $z_t$: $\psi_i \sim p_\theta(\psi|z_t, t)$ *(right)*. The model is trained on a 10 task split CIFAR-100 dataset. The separation of the latent samples and the task parameters of different models in the learned meta-space supports the usefulness of our method in overcoming catastrophic forgetting in continual learning.

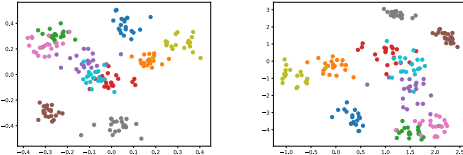

Figure 2: *(Left)* Plot of 20 samples drawn from each task-specific prior $z_t \sim p_\theta(z|t)$. *(Right)* t-SNE plot of weights obtained from decoder using corresponding $z_t$; $\psi_i \sim p_\theta(\psi|z_t, t)$. Each color corresponds to 10 diff tasks from split CIFAR100.

**5.4 Varying Number of Models for Ensembling:** In Sec 4, we sampled 30 models for each task from the decoder used in MERLIN. We

| # of models → | 1 | 30 | 50 | 100 |
|---|---|---|---|---|
| Split MNIST | 86.6 ± 1.4 | 90.6 ± 0.8 | 91.4 ± 0.3 | 91.6 ± 0.3 |
| Permuted MNIST | 79.7 ± 3.9 | 85.4 ± 0.4 | 85.6 ± 0.3 | 85.7 ± 0.2 |
| Split Mini-ImageNet | 30.1 ± 1.8 | 39.1 ± 2.1 | 40.5 ± 2.9 | 40.9 ± 1.3 |

Table 3: Test accuracy with varying number of models in our inference ensemble.

now vary the number of models in Tab 3. We observe a $3.98\%$, $6.34\%$ and $8.99\%$ improvement on Split-MNIST, Permuted-MNIST and Split Mini-Imagenet, when using 30 models for ensembling, over using only 1. The performance improvement however is not very significant on further increase beyond 30 models. This suggests that while ensembling is required, the number of models need not be very large. Improving our ensembling strategy (beyond majority voting) is a direction of our future work.

**5.5 Storage Requirements:** MERLIN requires only parameters of task-specific prior $p_{\boldsymbol{\theta}}(\boldsymbol{z}|\boldsymbol{t})$ and decoder of the VAE, at inference time. The encoder is required only if we need to continually learn further tasks. None of the components in the architecture grows with number of tasks, making MERLIN scalable. Quantitative analysis on the size of classifier and VAE (meta-model) is presented in Tab 4. We note that meta-model size is always smaller than classifier, and is $8\times$ smaller than the storage

| # of params in $\rightarrow$ | Clf | VAE |
|---|---|---|
| Split MNIST | 89610 | 31426 |
| Permuted MNIST | 89610 | 31446 |
| Split CIFAR-10 | 31307 | 17910 |
| Split CIFAR-100 | 34997 | 33810 |
| Mini-ImageNet | 34997 | 33810 |

Table 4: Num of params in classifier and VAE.

requirement of GSS, GEM, EWC and iCaRL which use ResNet-18 with 272,260 parameters. Similar to GEM, we maintain an exemplar buffer of 200 and 400 for MNIST and other datasets respectively.

**5.6 Varying Chunk Size:** As in Sec 4.1.4, we split the weight vector into equal-sized chunks before encoding it using VAE. We vary chunk size and report the average accuracy in Tab 5. As chunk size increases, the accuracy drops - we

| Chunk Size $\rightarrow$ | 100 | 300 | 1000 | 2000 |
|---|---|---|---|---|
| Split MNIST | $90.8 \pm 0.4$ | $90.6 \pm 0.8$ | $87.5 \pm 3.7$ | $87.4 \pm 2.2$ |
| Permuted MNIST | $85.6 \pm 0.4$ | $85.5 \pm 0.5$ | $83.7 \pm 2.8$ | $82.5 \pm 3.6$ |
| Split Mini-ImageNet | $40.9 \pm 1.2$ | $40.5 \pm 2.9$ | $38.8 \pm 2.8$ | $37.8 \pm 2.1$ |

Table 5: Avg test accuracy when varying chunk size.

hypothesize this is to do with the capacity of VAE. We use simple architectures in our encoder-decoder, and modeling larger weight chunks (or whole models) may require more complex VAE.

**5.7 Varying Exemplar Memory Size** We vary the memory size of the exemplar buffer for MERLIN, and compare the performance against our best competitors in Table 1 of the main paper: GEM [48] and iCaRL [63] in CIFAR-10 experiments. The results in Table 6 show that the increase in performance for MERLIN is sig-

| Memory Size $\rightarrow$ | 100 | 500 | 1000 | 2000 |
|---|---|---|---|---|
| GEM | $77.4 \pm 2.6$ | $79.9 \pm 1.9$ | $80.9 \pm 1.9$ | $80.5 \pm 1.5$ |
| iCaRL | $72.5 \pm 2.6$ | $73.6 \pm 1.3$ | $73.7 \pm 2.6$ | $74.8 \pm 1.3$ |
| MERLIN | $77.9 \pm 1.3$ | $81.9 \pm 0.3$ | $86.9 \pm 0.6$ | $88.4 \pm 0.8$ |

Table 6: Results of varying memory size of exemplar buffer for MERLIN, GEM [48] and iCaRL [63].

nificantly better when compared to GEM and iCaRL. We hypothesize this is because the weights sampled from our parameter distribution use these exemplars better, similar to how other meta-learning approaches like MAML [23] finetune on just a few samples. Using better exemplar selection methods other than random sampling (used in this work) can further enhance MERLIN, and is a direction of our future work.

## 6 Conclusion

We introduce MERLIN, a novel approach for online continual learning, based on consolidation in a meta-space of model parameters. Our method is modeled based on a VAE, however adapted to this problem through components such as task-specific learned priors. Our experimental evaluation on five standard continual learning benchmark datasets against five baseline methods brings out the efficacy of our approach. MERLIN can handle both class-incremental and domain-incremental settings, and can work with or without task information at test time. Understanding the dynamics of the latent space of the parameter distribution to further enhance memory retention and extending the methodology to a few-shot setting can be immediate follow-up efforts to MERLIN.

## Acknowledgements

We thank TCS for funding Joseph through its PhD fellowship, and DST, Govt of India, for partly supporting this work through the IMPRINT program (IMP/2019/000250). We also thank the members of Lab1055, IIT Hyderabad and the ContinualAI community for the engaging and fruitful discussions, which helped to shape MERLIN. Last but not the least, we thank all our anonymous reviewers for their insightful comments and suggestions, which helped to improve the quality and presentation of the paper.

## Broader Impact

Continual learning is a key desiderata for Artificial General Intelligence (AGI). Hence, this line of research has the benefits as well as the pitfalls of any other research effort geared in this direction. In particular, our work can help deliver impact on making smarter AI products and services, which can learn and update themselves on-the-fly when newer tasks and domains are encountered, without forgetting previously acquired knowledge. This is a necessity in any large-scale deployments of machine learning and computer vision, including in social media, e-commerce, surveillance, e-governance, etc - each of which have newer settings, tasks or domains added continually over time. Any negative effect of our work, such as legal and ethical concerns, are not unique to this work - to the best of our knowledge, but are shared with any other new development in machine learning, in general.

## Footnotes

[1] https://github.com/JosephKJ/merlin

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
