[Supplementary Material]

## Appendix: Meta-Consolidation for Continual Learning

In this appendix, we discuss the following details, which could not be included in the main paper owing to space constraints:

- Derivations of Equation 1 and Equation 2 in the main paper.
- Effect of adding an auxiliary classification loss while training the meta-model.
- A plot on the behavior of VAE training loss.
- A visualization of the latent space of the learned priors and the generated models.
- Further details of datasets used.
- Task-wise accuracies of the main results presented in Table 1.
- Connections of MERLIN to neuroscience literature.
- Comparison with Bayesian continual learning methods.
- Efficacy of task-specific learned priors.
- Results while using smaller backbones for the baselines.
- 5 class-per-task experiments.
- Our code implementation of our methodology in Section 3.

## A  Estimating Marginal Likelihood

Let $q_\phi(z|\psi,t)$ be the approximation for the intractable true posterior distribution $p_\theta(z|\psi,t)$ as defined in Section 3.1. The KL Divergence between these two distributions, $D_{KL}(q_\phi(z|\psi,t) \,||\, p_\theta(z|\psi,t))$ - written below as $D_{KL}(q\,||\,p)$ for convenience - can be expressed as follows:

$$
\begin{aligned}
D_{KL}(q\,||\,p) =\ & -\int_z q_\phi(z|\psi,t)\log\frac{p_\theta(z|\psi,t)}{q_\phi(z|\psi,t)} \\
=\ & -\int_z q_\phi(z|\psi,t)\log\frac{p_\theta(z|\psi,t)p_\theta(\psi|t)}{q_\phi(z|\psi,t)p_\theta(\psi|t)} \\
=\ & -\int_z q_\phi(z|\psi,t)\log\frac{p_\theta(z,\psi|t)}{q_\phi(z|\psi,t)p_\theta(\psi|t)} \\
=\ & -\int_z q_\phi(z|\psi,t)\log\frac{p_\theta(z,\psi|t)}{q_\phi(z|\psi,t)} + \log p_\theta(\psi|t)
\end{aligned}
$$

Rearranging, we get the marginal likelihood (Equation 1):

$$
\log p_\theta(\psi|t) =\ D_{KL}(q_\phi(z|\psi,t)\,||\,p_\theta(z|\psi,t)) + \underbrace{\int_z q_\phi(z|\psi,t)\ \log\frac{p_\theta(z,\psi|t)}{q_\phi(z|\psi,t)}}_{\mathcal{L}(\theta,\phi|\psi,t)}
$$

## B  Alternate form of ELBO

The evidence lower bound (ELBO), $\mathcal{L}(\theta,\phi|\psi,t)$ - written below as $\mathcal{L}$ for convenience - derived in the above section, can be rewritten as follows:

$$
\begin{aligned}
\mathcal{L} =\ & \int_z q_\phi(z|\psi,t)\ \log\frac{p_\theta(z,\psi|t)}{q_\phi(z|\psi,t)} \\
=\ & \mathbb{E}_{q_\phi(z|\psi,t)}[-\log q_\phi(z|\psi,t) + \log p_\theta(z,\psi|t)] \\
=\ & \mathbb{E}_{q_\phi(z|\psi,t)}[-\log q_\phi(z|\psi,t) + \log p_\theta(\psi|z,t) + \log p_\theta(z|t)] \\
=\ & \mathbb{E}_{q_\phi(z|\psi,t)}[-\log q_\phi(z|\psi,t) + \log p_\theta(z|t)]\ +\ \mathbb{E}_{q_\phi(z|\psi,t)}[\log p_\theta(\psi|z,t)] \\
=\ & \mathbb{E}_{q_\phi(z|\psi,t)}[\frac{\log p_\theta(z|t)}{\log q_\phi(z|\psi,t)}]\ +\ \mathbb{E}_{q_\phi(z|\psi,t)}[\log p_\theta(\psi|z,t)] \\
=\ & -D_{KL}(q_\phi(z|\psi,t)\,||\,p_\theta(z|t))\ +\ \mathbb{E}_{q_\phi(z|\psi,t)}[\log p_\theta(\psi|z,t)]
\end{aligned}
$$

This is the expression of ELBO used in Equation 2.

## C Minimizing ELBO and Classification Loss while Training VAE

The Evidence Lower-Bound (ELBO), defined in Equation 2, is maximized while training the VAE to learn the parameter distribution $p(\psi|t)$ for all the experiments reported so far. We now introduce one more loss term to guide the training of VAE and study its usefulness. The weights that are generated from the decoder $\psi \sim p_\theta(\psi|z_t, t)$, are used to initialize a classifier network. The loss for this network (computed on a training set of the classification dataset) is computed and also used to update the VAE.

| Datasets → | Split MNIST | | Permuted MNIST | | Split CIFAR-10 | | Split CIFAR-100 | | Split Mini-ImageNet | |
|---|---|---|---|---|---|---|---|---|---|---|
| Methods ↓ | A (↑) | F (↓) | A (↑) | F (↓) | A (↑) | F (↓) | A (↑) | F (↓) | A (↑) | F (↓) |
| ELBO | $90.7 \pm 0.8$ | $6.4 \pm 1.2$ | $85.5 \pm 0.5$ | $0.4 \pm 0.4$ | $82.9 \pm 1.2$ | $-0.9 \pm 1.9$ | $43.5 \pm 0.6$ | $2.9 \pm 3.7$ | $40.1 \pm 0.9$ | $2.8 \pm 3.2$ |
| ELBO + Clf Loss | $91.4 \pm 0.2$ | $6.1 \pm 0.6$ | $85.7 \pm 0.3$ | $0.8 \pm 0.1$ | $82.8 \pm 1.1$ | $0.2 \pm 1.3$ | $42.5 \pm 1.2$ | $-1.1 \pm 1.7$ | $40.8 \pm 2.6$ | $0.9 \pm 0.7$ |

Table 7: Average accuracy (A) and average forgetting measure (F) while training MERLIN with and without an auxiliary classification loss. All results in the main paper were obtained by optimizing only the Evidence Lower-Bound (ELBO) as defined in Equation 2, to which this classification loss is now added.

We run MERLIN on all datasets after training with this additional loss term, and the results are reported in Table 7. We see that this adds some improvement on certain datasets, but is in general marginal (except forgetting measure on CIFAR-100 and Mini-ImageNet, which shows good improvement). This generally implies that the VAE is able to capture the task solving information implicitly, without explicit global loss to enforce this. However, in more complex datasets such as ImageNet, adding a classification loss can further improve performance.

## D Loss Plots While Training VAE

We plot the loss curve while training the VAE in Figure 2. We observe that the VAE stabilizes fairly quickly, viz, within the first 10 epochs - further corroborating the usefulness of the proposed method. We plot the classification loss for completeness. The curve was plotted while training our method on the Permuted MNIST dataset.

Figure 2: Figure plots the ELBO, classification loss and the total loss against epochs. We see fair stability in the VAE training from the plot.

## E Visualization of Task-specific Priors and Sampled Models:

Figure 3: *(Left)* Plot of 20 samples drawn from each task-specific prior $z_t \sim p_\theta(z|t)$. *(Right)* t-SNE plot of weights obtained from decoder using corresponding $z_t$; $\psi_i \sim p_\theta(\psi|z_t, t)$. Each color corresponds to 10 diff tasks from split CIFAR100.

In Fig 3, we visualize the latent variables sampled from the learned prior distribution $z_t \sim p_\theta(z|t)$ *(left)* - which are two-dimensional themselves, and the 2D t-SNE embeddings [49] of their corresponding model parameters generated using $z_t$: $\psi_i \sim p_\theta(\psi|z_t, t)$ *(right)*. The model is trained on a 10 task split CIFAR-100 dataset. The separation of the latent samples and the task parameters of different models in the learned meta-space supports the usefulness of our method in overcoming catastrophic forgetting in continual learning.

# F   Dataset Description

As shown in Section 4 of the main paper, we evaluate MERLIN against multiple baselines on Split MNIST [13], Permuted MNIST [88], Split CIFAR-10 [88], Split CIFAR-100 [63] and Split Mini-ImageNet [15] datasets. Table 8 provides a summary of these benchmark datasets and the corresponding tasks involved, as used commonly in continual learning literature [15, 88, 63, 13] and in our work. The table enumerates different classes that form each task for class-incremental datasets.

| Dataset | # of images per task | Tasks |
|---|---|---|
| Split MNIST [13] | 1000 | $\tau_1 = \{0, 1\}$<br>$\tau_2 = \{2, 3\}$<br>$\tau_3 = \{4, 5\}$<br>$\tau_4 = \{6, 7\}$<br>$\tau_5 = \{8, 9\}$ |
| Permuted MNIST [88] | 1000 | 10 different spatial permutations, each corresponding to a task. |
| Split CIFAR-10 [88] | 2500 | $\tau_1 = \{$airplane, automobile$\}$<br>$\tau_2 = \{$bird, cat$\}$<br>$\tau_3 = \{$deer, dog$\}$<br>$\tau_4 = \{$frog, horse$\}$<br>$\tau_5 = \{$ship, truck$\}$ |
| Split CIFAR-100 [63] | 2500 | $\tau_1 = \{$beaver, dolphin, otter, seal, whale, aquarium fish, flatfish, ray, shark, trout$\}$<br>$\tau_2 = \{$orchids, poppies, roses, sunflowers, tulips, bottles, bowls, cans, cups, plates$\}$<br>$\tau_3 = \{$apples, mushrooms, oranges, pears, sweet peppers, clock, computer keyboard, lamp, telephone, television$\}$<br>$\tau_4 = \{$bed, chair, couch, table, wardrobe, bee, beetle, butterfly, caterpillar, cockroach$\}$<br>$\tau_5 = \{$bear, leopard, lion, tiger, wolf, bridge, castle, house, road, skyscraper$\}$<br>$\tau_6 = \{$cloud, forest, mountain, plain, sea, camel, cattle, chimpanzee, elephant, kangaroo$\}$<br>$\tau_7 = \{$fox, porcupine, possum, raccoon, skunk, crab, lobster, snail, spider, worm$\}$<br>$\tau_8 = \{$baby, boy, girl, man, woman, crocodile, dinosaur, lizard, snake, turtle$\}$<br>$\tau_9 = \{$hamster, mouse, rabbit, shrew, squirrel, maple, oak, palm, pine, willow$\}$<br>$\tau_{10} = \{$bicycle, bus, motorcycle, pickup truck, train, lawn-mower, rocket, streetcar, tank, tractor$\}$ |
| Split Mini-ImageNet [15] | 2500 | $\tau_1 = \{$goose, Ibizan hound, white wolf, mierkat, rhinoceros, beetle, cannon, carton, catamaran, combination lock, dustcart$\}$<br>$\tau_2 = \{$high bar, iPod, miniskirt, missile, poncho, coral reef, house finch, american robin, triceratops, green mamba$\}$<br>$\tau_3 = \{$daddy longlegs, toucan, jellyfish, dugong, walker hound, gazelle hound, gordon setter, komondor, boxer, tibetan mastiff$\}$<br>$\tau_4 = \{$french bulldog, newfoundland dog, miniature poodle, white fox, ladybug, three-toed sloth, rock beauty, aircraft carrier, ashcan, barrel$\}$<br>$\tau_5 = \{$beer bottle, carousel, chime, clog, cocktail shaker, dishrag, dome, file cabinet, fireguard, skillet$\}$<br>$\tau_6 = \{$hair slide, holster, lipstick, hautboy, pipe organ, parallel bars, pencil box, photocopier, prayer mat, reel$\}$<br>$\tau_7 = \{$slot, snorkel, solar dish, spider web, stage, tank, tile roof, tobacco shop, unicycle, upright piano$\}$<br>$\tau_8 = \{$wok, worm fence, yawl, street sign, consomme, hotdog, orange, cliff, mushroom, capitulum$\}$<br>$\tau_9 = \{$nematode, king crab, golden retriever, malamute, dalmatian, hyena dog, lion, ant, ferret, bookshop$\}$<br>$\tau_{10} = \{$crate, cuirass, electric guitar, hourglass, mixing bowl, school bus, scoreboard, theater curtain, vase, trifle$\}$ |

Table 8: Summary of benchmark datasets used in continual learning literature [15, 88, 63, 13] and this work. The number of images that form a task and the classes in each task for the class-incremental setting are shown.

# G   Task-wise Accuracy

Tables 9 through Table 13 above show task-wise accuracy on the various continual learning datasets used in this work. The average of these accuracy values is presented in Table 1 of the main paper.

The tasks defined on continual versions of CIFAR and ImageNet datasets have implicit semantic meaning. On Split CIFAR-100, there is a significant semantic shift when adding $\tau_5$, which includes animals (`bear`, `leopard`, `lion`, `tiger`, `wolf`) and man-made structures (`bridge`, `castle`, `house`, `road`, `skyscraper`). So far the model has been trained on fishes, flowers, fruits and insects. In this challenging setting, MERLIN achieves an improvement of $4.61\%$ in accuracy. Accuracy of GSS and EWC falls while iCaRL and GEM has minor improvement of $1.83\%$ and $1.51\%$.

In Split CIFAR-10, while adding `deer` and `dog` class ($\tau_3$) to the model that is already trained on `bird` and `cat`, the accuracy of all the methods fall, while MERLIN improves the accuracy by $0.25\%$. We observe an improvement of $2.80\%$ when we add `ship` and `truck` as part of $\tau_5$. The performance of GSS and GEM decreases, while iCaRL, which does well even in Table 1 in the main paper, improves ($2.38\%$) similar to MERLIN. This setting is challenging because the previous three tasks were only animal classes, and there is a semantic shift at $\tau_5$. The grouping of classes in Split Mini-ImageNet is more or less random. Here, we note that in four out of ten tasks ($\tau_5, \tau_7, \tau_8, \tau_9$), MERLIN achieves better improvement in accuracy when compared to the other methods. On Split CIFAR-100 and Split Mini-ImageNet, GSS degenerates (even after carefully choosing the hyper-parameters) as the number of tasks increase. We note that GSS was not evaluated on these datasets in the original paper.

| Tasks → | $\tau_1$ | $\tau_2$ | $\tau_3$ | $\tau_4$ | $\tau_5$ |
|---|---|---|---|---|---|
| Single | 99.9 ± 0.1 | 47.9 ± 0.8 | 32.9 ± 0.2 | 24.8 ± 0 | 19 ± 0.3 |
| EWC | 99.9 ± 0.1 | 48.2 ± 0.3 | 32.8 ± 0.2 | 24.8 ± 0 | 19.3 ± 0.1 |
| GEM | 99.8 ± 0.1 | 92.1 ± 0.8 | 85.1 ± 3.1 | 82 ± 2.5 | 75.2 ± 1.4 |
| iCaRL | 99.7 ± 0.1 | 96.3 ± 1.0 | 94.9 ± 1.1 | 83.1 ± 1.0 | 75.6 ± 1.4 |
| GSS | 99.9 ± 0.1 | 96.2 ± 0.3 | 90.9 ± 0.9 | 84.6 ± 0.9 | 70.4 ± 1.9 |
| MERLIN | 99.2 ± 0.1 | 93.3 ± 0.7 | 88.7 ± 0.3 | 88.6 ± 1.6 | 83.4 ± 1.2 |

Table 9: Task wise accuracy on Split MNIST dataset.

| Tasks → | $\tau_1$ | $\tau_2$ | $\tau_3$ | $\tau_4$ | $\tau_5$ |
|---|---|---|---|---|---|
| Single | 86.9 ± 0.6 | 73.5 ± 3.4 | 68 ± 4.6 | 68 ± 3.6 | 69.9 ± 3.2 |
| EWC | 85.8 ± 1.7 | 73.9 ± 1.3 | 71.7 ± 3.5 | 70.3 ± 1.4 | 69.7 ± 3 |
| GEM | 84.4 ± 1.2 | 78.3 ± 1.6 | 76.6 ± 2.9 | 78.6 ± 0.7 | 77.8 ± 2 |
| iCaRL | 87.2 ± 0.9 | 73.7 ± 2.1 | 69.5 ± 0.4 | 65.3 ± 2.1 | 67.7 ± 1.1 |
| GSS | 90 ± 0.3 | 66.3 ± 4.8 | 51.8 ± 1.7 | 45 ± 2.5 | 36.5 ± 4 |
| MERLIN | 87.6 ± 0.7 | 80.2 ± 1.2 | 80.5 ± 1.4 | 81.8 ± 1 | 84.6 ± 1.5 |

Table 10: Task wise acc on Split CIFAR-10 dataset.

| Tasks → | $\tau_1$ | $\tau_2$ | $\tau_3$ | $\tau_4$ | $\tau_5$ | $\tau_6$ | $\tau_7$ | $\tau_8$ | $\tau_9$ | $\tau_{10}$ |
|---|---|---|---|---|---|---|---|---|---|---|
| Single | 79.2 ± 4.3 | 79 ± 1.2 | 77.9 ± 2.9 | 74 ± 1.8 | 76.7 ± 1.1 | 72.9 ± 2.6 | 69 ± 1 | 68.5 ± 3.4 | 68.4 ± 3.1 | 65.7 ± 1.4 |
| EWC | 79.1 ± 5.3 | 78.8 ± 1.5 | 78.4 ± 2.5 | 75.2 ± 1.4 | 77.7 ± 0.5 | 75.4 ± 1.4 | 72.7 ± 2.6 | 71.9 ± 2.1 | 71.5 ± 0.8 | 69 ± 2.3 |
| GEM | 77.7 ± 14.2 | 81.5 ± 13.3 | 82.3 ± 1.8 | 80 ± 11.9 | 82.9 ± 1.5 | 83.6 ± 0.1 | 82.8 ± 5.8 | 83.4 ± 0.1 | 83.2 ± 0.4 | 83.2 ± 0.4 |
| GSS | 86 ± 1.2 | 84.9 ± 1.2 | 83.9 ± 1.5 | 83.3 ± 1.2 | 82.5 ± 1.2 | 81.7 ± 0.5 | 79 ± 1.7 | 79 ± 1.7 | 78 ± 0.9 | 75.9 ± 1.6 |
| MERLIN | 85.8 ± 0.4 | 85.3 ± 0.6 | 85 ± 0.6 | 85.7 ± 0.7 | 85.5 ± 0.4 | 85.3 ± 0.7 | 85.8 ± 0.4 | 85.5 ± 0.4 | 85.6 ± 0.5 | 85.8 ± 0.2 |

Table 11: Task wise accuracy on Permuted MNIST dataset.

| Tasks → | $\tau_1$ | $\tau_2$ | $\tau_3$ | $\tau_4$ | $\tau_5$ | $\tau_6$ | $\tau_7$ | $\tau_8$ | $\tau_9$ | $\tau_{10}$ |
|---|---|---|---|---|---|---|---|---|---|---|
| Single | 40 ± 5.7 | 32.2 ± 3.9 | 29 ± 8.1 | 31.1 ± 2.9 | 29.8 ± 2 | 30.1 ± 1.2 | 29.8 ± 4.1 | 28.9 ± 3.3 | 29.3 ± 2.5 | 28 ± 2.1 |
| EWC | 37.1 ± 3.5 | 32.6 ± 2.3 | 25 ± 9.4 | 30.4 ± 2 | 30.2 ± 2.2 | 28.8 ± 4.4 | 26.7 ± 4 | 28.4 ± 1.5 | 28 ± 1.5 | 25.1 ± 3 |
| GEM | 30.9 ± 3 | 32.9 ± 1.1 | 35.5 ± 2.8 | 39.3 ± 2.1 | 40.8 ± 2.1 | 41.8 ± 2.3 | 44.8 ± 1.7 | 46.2 ± 1.3 | 46.6 ± 0.8 | 47.6 ± 2.4 |
| iCaRL | 22.7 ± 1.3 | 20.8 ± 4 | 22.2 ± 2.7 | 24.5 ± 4.2 | 26.4 ± 3.5 | 27.2 ± 3.4 | 30.3 ± 3.1 | 31.5 ± 2.8 | 31.9 ± 2.2 | 33.9 ± 2.6 |
| GSS | 51 ± 1.9 | 31.8 ± 1.8 | 25.7 ± 0.5 | 17.8 ± 0.6 | 15.2 ± 0.4 | 12.5 ± 0.6 | 11.2 ± 0.3 | 9.6 ± 0.3 | 8.6 ± 0.4 | 8.3 ± 0.2 |
| MERLIN | 47.6 ± 2.1 | 37.6 ± 2.1 | 38.7 ± 1.1 | 40.1 ± 2.5 | 44.7 ± 0.9 | 43.1 ± 2.1 | 44.8 ± 2.8 | 47.2 ± 2.6 | 45.5 ± 2.5 | 46.1 ± 3.7 |

Table 12: Task wise accuracy on Split CIFAR-100 dataset.

| Tasks → | $\tau_1$ | $\tau_2$ | $\tau_3$ | $\tau_4$ | $\tau_5$ | $\tau_6$ | $\tau_7$ | $\tau_8$ | $\tau_9$ | $\tau_{10}$ |
|---|---|---|---|---|---|---|---|---|---|---|
| Single | 40.8 ± 2.5 | 30.8 ± 1.4 | 29.5 ± 1.5 | 26.4 ± 4.3 | 26.9 ± 3.5 | 24.8 ± 2.3 | 25.8 ± 2.3 | 24 ± 3.4 | 23.2 ± 3.3 | 23.5 ± 1.8 |
| EWC | 41.8 ± 2.4 | 30.8 ± 3.1 | 29.1 ± 3.3 | 28.4 ± 1.5 | 27.3 ± 2.2 | 24.7 ± 2.8 | 26.1 ± 1.9 | 25.1 ± 1.1 | 23.5 ± 3.3 | 23.2 ± 4.3 |
| GEM | 39.2 ± 3 | 38.1 ± 2.2 | 38.1 ± 2.3 | 39.9 ± 2.3 | 39.8 ± 1 | 39.1 ± 1.2 | 38.3 ± 1.5 | 38.5 ± 1.6 | 38.9 ± 1 | 38.8 ± 0.4 |
| iCaRL | 41.8 ± 1 | 30.4 ± 2 | 33.9 ± 0.4 | 33.2 ± 0.9 | 34.8 ± 1.7 | 33.1 ± 0.7 | 34.8 ± 1.4 | 32.5 ± 1.3 | 32 ± 1.7 | 35.2 ± 1.1 |
| GSS | 44.6 ± 2.1 | 26.1 ± 2.4 | 17.5 ± 1.3 | 15 ± 1.3 | 11.1 ± 0.8 | 8.5 ± 0.7 | 7.9 ± 0.5 | 6.5 ± 0.6 | 5.6 ± 0.6 | 5.3 ± 0.7 |
| MERLIN | 46.5 ± 1.9 | 38 ± 1.9 | 36.5 ± 0.3 | 34.7 ± 1.7 | 40.5 ± 3.7 | 37.6 ± 4.7 | 41 ± 3.9 | 42.1 ± 4.6 | 41.8 ± 3.6 | 41.7 ± 3 |

Table 13: Task wise accuracy on Split Mini-ImageNet dataset.

# H  Connections to Neuroscience Literature

Wilson and McNaughton, in their seminal paper [85] made the following observation: "...initial storage of event memory occurs through rapid synaptic modification, primarily within the hippocampus. During subsequent slow-wave sleep, synaptic modification within the hippocampus itself is suppressed and the neuronal states encoded within the hippocampus are "played back" as part of a consolidation process by which hippocampal information is gradually transferred to, the neocortex." Further, other work [33, 86, 5, 76] extended the connection between memory consolidation and improved continual learning capabilities. One can view the phases in our method mirroring the above observation. The base network training (Figure 1, left) is similar to the learning that happens in the hippocampus. After learning a new task, we have a period of inactivity (similar to sleep), where the model weights are encoded, the parameter distribution is re-learned and consolidated by "playing back" model parameters from all learned tasks (Figure 1, middle). Interestingly, many years ago, [65] used auto-associative neural networks [40] to study connections between sleep-induced consolidation and reduction in catastrophic forgetting. Recently, [26] studied the same on a custom build biophysically-realistic thalamocortical network model. The encouraging results from such studies adds further motivation to our proposed methodology.

# I  Comparison with Bayesian Continual Learning Methods

We compare with a recent bayesian continual learing method, CN-DPM [44], for completeness of our work. We report the results in Tab 14. As shown in the table, CN-DPM performs better than MERLIN on Split-MNIST, but drastically fails on harder datasets. We note that the baseline methods considered in this work also perform better than CN-DPM on non-MNIST datasets.

| Methods | Split MNIST | Permuted MNIST | Split CIFAR10 | Split CIFAR100 | Mini-ImageNet |
|---|---|---|---|---|---|
| Single | 44.89 ± 0.30 | 73.13 ± 2.27 | 73.24 ± 3.08 | 30.81 ± 3.57 | 27.57 ± 2.64 |
| EWC | 45.01 ± 0.14 | 74.98 ± 2.04 | 74.28 ± 2.2 | 29.23 ± 3.38 | 28 ± 2.59 |
| GEM | 86.79 ± 1.56 | 82.05 ± 4.95 | 79.13 ± 1.68 | 40.65 ± 1.95 | 34.17 ± 1.23 |
| iCaRL | 89.91 ± 0.92 | NA | 72.65 ± 1.33 | 27.13 ± 2.99 | 38.86 ± 1.63 |
| GSS | 88.39 ± 0.81 | 81.44 ± 1.27 | 57.9 ± 2.65 | 19.19 ± 0.7 | 14.81 ± 0.98 |
| MERLIN | 90.67 ± 0.80 | 85.54 ± 0.5 | 82.93 ± 1.16 | 43.55 ± 0.61 | 40.05 ± 2.94 |
| CN-DPM [44] | 92.12 ± 0.14 | - | 46.01 ± 1.23 | 14.29 ± 0.14 | - |
| MERLIN - SN Prior | 23.34 ± 0.24 | 32.51 ± 1.57 | 28.23 ± 2.21 | 12.32 ± 1.45 | 14.76 ± 0.23 |

Table 14: Comparison with CN-DPM & MERLIN using Standard Normal (SN) Prior.

Not all VAE-based continual learning methods learn a posterior over model parameters, or operate in an online continual learning setting. For eg., VCL [56] learns a posterior over the data distribution (also not an online continual learning method), and not model parameter distribution. This is a subtle difference to be noted. MERLIN performs variational continual learning in the model parameter space, can be adapted easily to class and domain incremental setting, and work in task-aware or task-agnostic settings.

## J   Efficacy of Task-specific Learned Priors

The learned task-specific priors are necessary to generate task-specific weights and consolidate meta-model on previous task parameters, as well as to sample models for ensembling at inference. We ran a study where we replaced the task-specific learned prior with a standard Normal prior and finetuned the corresponding generated model on task-specific exemplars. We showcase the results in Tab 14, where we see that MERLIN fails drastically, validating the usefulness of task-specific learned priors.

## K   Results on Additional Datasets

Along with the standard continual learning benchmarks that are reported in the main paper, we additionally compare with Heterogeneous dataset introduced by Serra *et al.* [72]. We also explore continually learning in the audio modality with Audio MNIST dataset [7]. We comfortably outperform baselines on both these datasets too.

| Methods | HAT[72] | AudioMNIST[7] |
|---|---|---|
| Single | $47.79 \pm 0.94$ | $76.37 \pm 2.25$ |
| EWC | $47.19 \pm 2.58$ | $79.89 \pm 16.14$ |
| GEM | $67.23 \pm 0.97$ | $89.45 \pm 1.14$ |
| iCaRL | $62.34 \pm 2.45$ | $84.73 \pm 2.22$ |
| GSS | $69.79 \pm 1.51$ | $92.81 \pm 0.19$ |
| MERLIN | $\mathbf{73.54 \pm 1.71}$ | $\mathbf{96.47 \pm 1.79}$ |

Table 15: Results on heterogeneous dataset from HAT[72] and AudioMNIST[7].

## L   Using Smaller Backbones for the Baselines

MERLIN uses smaller backbones for the non-MNIST experiments. For the results in the main paper, we use larger backbones for the baseline, which is unfair for MERLIN, which still outperforms them. We re-ran all baselines with the same smaller ResNet used in MERLIN, and report the results in Tab 16. We see that MERLIN outperform all baselines here again; the performance of the baseline model drops significantly, possibly due to the smaller capacity.

| Methods | Split CIFAR10 | Split CIFAR100 | Mini-ImageNet |
|---|---|---|---|
| Single | $69.65 \pm 0.79$ | $18.8 \pm 2.21$ | $18.57 \pm 2.31$ |
| EWC | $67.98 \pm 2.96$ | $16.89 \pm 3.95$ | $19.29 \pm 3.58$ |
| GEM | $72.23 \pm 1.56$ | $26.71 \pm 1.75$ | $27.71 \pm 2.56$ |
| iCaRL | $69.23 \pm 2.24$ | $24.81 \pm 2.88$ | $23.84 \pm 1.95$ |
| GSS | $49.82 \pm 2.01$ | $13.99 \pm 0.56$ | $12.92 \pm 0.17$ |
| MERLIN | $\mathbf{82.93 \pm 1.16}$ | $\mathbf{43.55 \pm 0.61}$ | $\mathbf{40.05 \pm 2.94}$ |

Table 16: Comparison with baselines with smaller ResNet backbone

## M   5 class-per-task Experiments

Experiments on CIFAR-100 and Mini-ImageNet datasets in the main paper uses 10 class-per-task. Here we run the incremental experiments with 5 class-per-task. The results are reported in Tab 17. We perform better than baselines even in this setting.

| Methods | Split CIFAR100 | Mini-ImageNet |
|---|---|---|
| Single | $36.44 \pm 3.44$ | $35.85 \pm 2.08$ |
| EWC | $37.03 \pm 2.51$ | $35.36 \pm 2.07$ |
| GEM | $57.02 \pm 1.41$ | $52.28 \pm 1.53$ |
| iCaRL | $50.23 \pm 1.37$ | $53.22 \pm 1.56$ |
| GSS | $18.74 \pm 0.82$ | $16.34 \pm 0.12$ |
| MERLIN | $\mathbf{64.83 \pm 1.78}$ | $\mathbf{57.35 \pm 1.92}$ |

Table 17: Accuracy values when MERLIN incrementally learn 20 tasks on Split CIFAR-100 and Mini-Imagenet Datasets.

## N   Code

We share the code for MERLIN here: `https://github.com/JosephKJ/merlin`