[Reviews · NeurIPS 2020]

Review 1

Summary and Contributions: The paper proposes an online continual learning method, MERLIN, that learns a distribution over the task-specific model parameters given a context (task identifiers, etc). VAE is used to model the distribution over the model parameters. More specifically, given a dataset of a task t, the idea is to train ‘B’ separate models. A VAE is then trained, using these ‘B’ model parameters as training points to learn an encoder (mapping the parameters to the latent) and decoder (mapping the latent to model parameters). The standard VAE ELBO is maximized during the training. One notable change is that the (parametric) prior over the latent distribution is task-specific and is learned along with the VAE parameters. After training each task, the updated VAE is consolidated for previous tasks by sampling from the task-specific learned priors, generating the parameters from those samples, and updating all the VAE parameters using those generated samples as supervisory signals. At inference time, the latent is sampled from the task-specific prior or all the priors (depending on whether the task information is present or not), ‘E’ number of models set is sampled from the decoder. This set is then fine-tuned on the replay buffer and the results are ensembled over the set. The experiments are reported on the standard continual learning benchmarks for image classification. ------------------------- Post-rebuttal: - The authors adequately addressed some of my concerns. While I still believe that Bayesian Continual Learning type baselines would be better suitable for this work and I encourage the authors to add those in their final draft, comparison with CN-DPM, if done correctly, suggests that MERLIN can outperform other Bayesian baselines (although I am not sure if the authors used CN-DPM correctly or in the right setting). I don't agree with the author's' assertion that VCL does not learn a distribution over model parameters. Anyhow, the rebuttal is strong and addressed most of my concerns. Therefore, I am increasing my score to marginally above acceptance threshold. I am not giving a clear accept because I still believe that the method is unnecessarily cumbersome and some of the components can be simplified.

Strengths: Positives: - By and large, the paper is well-written. Although the method seems overly complicated (more on this in the negative section) but the overall writing of the paper is very good. - Barring Bayesian continual learning, the paper is well-grounded in the recent literature.

Weaknesses: Negatives: 1) Why not posterior over model parameters: It’s not clear to me what is the advantage of this framework over standard variational continual learning type approaches (https://arxiv.org/abs/1710.10628, https://openreview.net/pdf?id=SJxSOJStPr, etc). In this work and in the VCL type approaches the objective is to model the distribution over network parameters. Could the authors point out why using model parameters as training data for VAE (like they did) is better than standard VAE training in a continual setting? It seems like a lot of machinery has been used in this work without properly grounding the study in the literature. The best baselines to study this work would have been VCL and likes. There are no comparisons with them in the experiments section. 2) Task-specific learned priors: Why do the authors choose these? What would happen if you take a standard normal prior or any other sensible prior and finetune the decoder on the exemplars? I can’t seem to find the experiments where this ablation has been done, which, frankly, makes the choice rather ad-hoc and unnecessarily cumbersome. You would want to experimentally show that this choice makes sense. 3) Experiments: There are a few issues with experiments. 3.1) For SplitCIFAR100 and miniImageNet, 10 classes per task would correspond to the 5000 samples per class and not the 2500 samples. The authors of the cited works used 5 samples per task, making 2500 samples per class. 3.2) Network architectures not being the same: It’s the standard practice in continual learning to make the network architectures the same size. More overly parameterized architectures are more prone to prune forgetting. The use of bigger architectures in the baselines may already hurt their performance. Please make a fair comparison using an equal number of parameters. 3.3) Baselines in the task-aware setting: Some of the baselines (GEM etc) are known to work well in the task-aware setting. The authors didn’t report the numbers of these baselines in that setting in the main paper. Please report that. Also, there is no information on the amount of episodic memory used for these baselines. Please provide that information. 3.4) As pointed out earlier, the best baselines to compare this work against are the ones based on Bayesian continual learning (VCL, etc). 4) Efficiency: The authors claim that at inference time their method works in real-time which seems strange given the number of steps they perform at inference time (Alg 3). Could you please provide the timing comparison of train/ test times with the baselines?

Correctness: See the weakness section.

Clarity: Yes.

Relation to Prior Work: See the weakness section.

Reproducibility: Yes

Additional Feedback:


Review 2

Summary and Contributions: The paper introduces a novel method for continual learning which can be applied both on task-free and task-aware scenario. It trains a network for prediction and a variational auto-encoder to learn latent space for encountered tasks hence it enables for generating the appropriate model parameters according to the tasks (meta-consolidation). At inference, the learned decoder generates the model parameters and reconstructs the network with the coreset which are randomly sampled from previous tasks. By aggregating prior for encountered tasks, the method generate the model parameter well.

Strengths: As they perform learning latent space across arriving tasks through a variational auto-encoder, the method successfully reconstruct task-specific knowledge at inference time with only a few of the coreset instances without occupying task-specific parameters. Also, the method can be applied on task-free continual learning scenario which are further realistic direction to the area.

Weaknesses: However, I'm not sure that how the latent space describe the task well for complex tasks or larger number of tasks. While the method is validated across various datasets, such as MNIST-variants and CIFAR-variants. I strongly recommend to do experiment and analyze the method for larger or multiple heterogeneous datasets from HAT [1]. Also, the authors assign too small margins for the paper, and it largely reduces a readability. [1] Serra, Joan, et al. "Overcoming catastrophic forgetting with hard attention to the task." ICML 2018.

Correctness: The method looks correct.

Clarity: The paper is written well and properly describes their methods and experimental procedures.

Relation to Prior Work: The method looks clearly differentiate with prior works.

Reproducibility: Yes

Additional Feedback: The paper shows interesting idea and outstanding performance compared to strong baselines. But the validation performed on conventional but simple datasets, and the margins between texts, pages, tables are too small that harms the quality of the paper. ======== Post-rebuttal: After I read other reviews and the author response with additional experiments, I believe that the work deserves to be accepted. I raise the score to 7.


Review 3

Summary and Contributions: In this paper, the authors propose a novel approach, which employs the consolidation in a meta-space of model parameters. The proposed method can handle both class-incremental and domain-incremental settings, and can work with or without task information at test time. -----------------------Final Decision The athuors have solved most of my concern. But I still think the strategy of the vote seems like a trick to improve the performance. Thus, I decide to keep my score unchanged.

Strengths: 1, The writing of this paper is clear and easy to understand. 2, The performance of the proposed method is impressive. 3, The experiments are sufficient to prove the effectiveness of the proposed method.

Weaknesses: 1,The scale of subsets for training basic models is not mentioned and discussed in the paper, which is an important part for the method in my opinion. 2, The vote of basic model seems like a trick to improve the performance. When the number is set as 1, the performance is not better than other methods.

Correctness: yes

Clarity: yes

Relation to Prior Work: yes

Reproducibility: Yes

Additional Feedback:


Review 4

Summary and Contributions: This paper outlines a novel method for continual learning in deep neural networks. The authors model the meta distribution of the model parameters, and proposes a variational auto-encoder (VAE) method to continually learn in this parameter space while doing model consolidation. This method of continual learning in the parameter space allows this approach to work in a task-aware setting when appropriate, and it operates in an online setting where only a single pass is made over the training data. The authors state that this is the first known method of incremental learning in the parameter space, to the best of their knowledge. After author feedback: Our initial recommendation of marginal accept of the paper remains unchanged.

Strengths: (1) The meta-learning approach adapts to new tasks arriving over time, and consolidates the tasks over time. (2) It models task-specific parameter distributions and does a meta-consolidation as necessary. (3) It does inference in both task-aware and task-agnostic modes, as necessary. (4) Experiments and user studies showed the effectiveness of the proposed approach called MERLIN.

Weaknesses: (1) No discussion is provided about why the forgetting measure is worse for MERLIN on Split MERLIN and Split CIFAR-100. (2) The results can be validated on more domains -- all the current results are shown on the image analysis domain.

Correctness: Yes

Clarity: Yes

Relation to Prior Work: Yes

Reproducibility: No

Additional Feedback:

[Author Response · NeurIPS 2020]

We thank all reviewers for the insightful feedback. We are encouraged to note that our method, MERLIN, is novel (**R2**,**R3**,**R5**); our experimental setting captures the method's effectiveness (**R3**,**R5**); we achieve 'outstanding performance compared to strong baselines'(**R2**); our approach is 'well-grounded' (**R1**), 'clearly different from prior works' (**R2**). **R2** notes that the task-free scenario used is a 'realistic direction to the area'. Further, all reviewers unanimously agree that 'the paper is well-written'. We address all comments below. We will use the additional page allowed in the final version to incorporate feedback and address any lack of clarity in presentation (**R2**).

(**R1**) **"Comparison with Bayesian Continual Leaning (BCL) methods":** Thanks for this question, it gives more completeness to our work. We compared MERLIN against the most recent BCL work, CN-DPM (Lee et al., ICLR'20) as **R1** suggested. As shown in the table, CN-DPM performs better than MERLIN on Split-MNIST, but drastically fails on harder datasets. We note that the baseline methods considered in this work

| Methods | Split MNIST | Permuted MNIST | Split CIFAR10 | Split CIFAR100 | Mini-ImageNet |
|---|---|---|---|---|---|
| Single | 44.89 ± 0.30 | 73.13 ± 2.27 | 73.24 ± 3.08 | 30.81 ± 3.57 | 27.57 ± 2.64 |
| EWC | 45.01 ± 0.14 | 74.98 ± 2.04 | 74.28 ± 2.2 | 29.23 ± 3.38 | 28 ± 2.59 |
| GEM | 86.79 ± 1.56 | 82.05 ± 4.95 | 79.13 ± 1.68 | 40.65 ± 1.95 | 34.17 ± 1.23 |
| iCaRL | 89.91 ± 0.92 | NA | 72.65 ± 1.33 | 27.13 ± 2.99 | 38.86 ± 1.63 |
| GSS | 88.39 ± 0.81 | 81.44 ± 1.27 | 57.9 ± 2.65 | 19.19 ± 0.7 | 14.81 ± 0.98 |
| MERLIN | 90.67 ± 0.80 | **85.54 ± 0.5** | **82.93 ± 1.16** | **43.55 ± 0.61** | **40.05 ± 2.94** |
| CN-DPM (ICLR'20) | **92.12 ± 0.14** | - | 46.01 ± 1.23 | 14.29 ± 0.14 | - |
| MERLIN - SN Prior | 23.34 ± 0.24 | 32.51 ± 1.57 | 28.23 ± 2.21 | 12.32 ± 1.45 | 14.76 ± 0.23 |

also perform better than CN-DPM on non-MNIST datasets. We'd also like to add that not all VAE-based CL methods learn a posterior over model params, or operate in an online CL setting. For eg., VCL (Nguyen et al, ICLR'18) learns a posterior over the data distribution (also not an online CL method), and not model param distribution. This is a subtle difference to be noted. MERLIN performs variational CL in the model param space, can be adapted easily to class + domain incremental setting, and work in task-aware + task-agnostic settings.

(**R1**) **"Why task-specific learned priors, not std normal prior?":** As correctly noted in **R1**'s 'Summary', the learned task-specific priors are necessary to generate task-specific weights and consolidate meta-model on previous task params, as well as to sample models for ensembling at inference. As suggested, we ran a study where we replaced the task-specific learned prior with a standard Normal prior and finetuned the corresponding generated model on task-specific exemplars. The last row of table above (MERLIN - SN Prior) shows the result (very poor), validating the usefulness of task-specific learned priors. We also visualized the learned-task specific prior in Fig 3 (Appendix), where we see good separability across tasks.

| Methods | HAT | AudioMNIST |
|---|---|---|
| Single | 47.79 ± 0.94 | 76.37 ± 2.25 |
| EWC | 47.19 ± 2.58 | 79.89 ±16.14 |
| GEM | 67.23 ± 0.97 | 89.45 ± 1.14 |
| iCaRL | 62.34 ± 2.45 | 84.73 ± 2.22 |
| GSS | 69.79 ± 1.51 | 92.81 ± 0.19 |
| MERLIN | **73.54 ± 1.71** | **96.47 ± 1.79** |

(**R2**, **R5**)**"Expts on heterogeneous datasets from HAT?; Results on more domains?":** Thanks for the suggestion. We ran expts on the Heterogeneous dataset from HAT as well as the AudioMNIST dataset (Becker et al, 2018) (shown in adjoining table). We comfortably outperform baselines on these datasets and domains too.

| Methods | Split CIFAR10 | Split CIFAR100 | Mini-ImageNet |
|---|---|---|---|
| Single | 69.65 ± 0.79 | 18.8 ± 2.21 | 18.57 ± 2.31 |
| EWC | 67.98 ± 2.96 | 16.89 ± 3.95 | 19.29 ± 3.58 |
| GEM | 72.23 ± 1.56 | 26.71 ± 1.75 | 27.71 ± 2.56 |
| iCaRL | 69.23 ± 2.24 | 24.81 ± 2.88 | 23.84 ± 1.95 |
| GSS | 49.82 ± 2.01 | 13.99 ± 0.56 | 12.92 ± 0.17 |
| MERLIN | **82.93 ± 1.16** | **43.55 ± 0.61** | **40.05 ± 2.94** |

(**R1**) **"Use of bigger architectures in baselines may hurt their performance":** We re-ran all baselines with the same smaller ResNet used in MERLIN (L248), and report the results in adjoining table. We see that MERLIN outperform all baselines here again; the performance of the baseline model drops significantly, possibly due to the smaller capacity.

(**R1**) **"For CIFAR100/miniImageNet, 10 classes/task corresponds to 5000 samples/class and not 2500? Cited works used 5 classes/task":** For these datasets, we randomly sampled 2500 samples from 5000, and used the same 2500 across all baselines, for fair comparison (results reported across 5 such trials). To further clarify, we ran experiments with 5 samples per task (20 tasks) and report the results in adjoining table. We note that baseline accuracy matches with values reported in GEM (Tab 2, Col 3). We perform better than baselines even in this setting.

| Methods | Split CIFAR100 | Mini-ImageNet |
|---|---|---|
| Single | 36.44 ± 3.44 | 35.85 ± 2.08 |
| EWC | 37.03 ± 2.51 | 35.36 ± 2.07 |
| GEM | 57.02 ± 1.41 | 52.28 ± 1.53 |
| iCaRL | 50.23 ± 1.37 | 53.22 ± 1.56 |
| GSS | 18.74 ± 0.82 | 16.34 ± 0.12 |
| MERLIN | **64.83 ± 1.78** | **57.35 ± 1.92** |

**Other clarifications:** - (**R3**) *"Scale of subsets":* Sec 4.1.1 has these details. The base model is trained with 1000 (MNIST) and 2500 (all datasets other than MNIST) samples per task; - (**R1**,**R3**) *"Amount of episodic memory used for baselines":* All baselines had access to same amount of exemplar memory as in MERLIN: 100 for MNIST and 600 for all other datasets. In Appendix Sec C, we study effect of varying exemplar memory size of MERLIN and two of its best competitors (GEM and iCaRL); - (**R1**) *"Inference time":* MERLIN take 745 ms while baseline methods take $\sim$ 300ms for CIFAR datasets on a single 1080Ti GPU. It takes more time than baselines, but is still real-time; - (**R5**) *"Why forgetting measure is worse for MERLIN on two datasets.":* No method is perfect. As discussed in L308-312, iCaRL uses distillation loss to ensure that logits of previous task don't alter much while learning a new task. This brings down the forgetting measure. We still outdo all baselines on 3 other datasets; - (**R1**) *"Baselines in task-aware setting":* All reported results of baselines in the paper are task-aware. Task-agnostic MERLIN was put under disadvantage while being compared to task-aware counterparts, still outperforming them. We believe that this confusion arose because of L 331, which should have been "All results *of MERLIN* in Tab 1 do not assume task information"; - (**R3**) *"vote of basic model...trick to improve performance":* Our method design allows ensembling of models for CL, though each model may be weak by itself (i.e. number of models is 1) - each individual model is upto $8\times$ smaller in param size than baseline models (L 353). Such an ensembling approach has not been tried before, and cannot be done easily with existing methods too.

[Meta-Review · NeurIPS 2020]

This work provides an interesting approach for continual learning over time that is able to work in both task-specific and task-agnostic scenarios. The paper is well written and easy to read. The author response addressed some of the reviewer's concerns and led to the decision to accept assuming the comparisons with Bayesian continual learning and same-size architectures are included in the final paper. The work could be improved by adding more heterogenous and complex tasks or by extending to domains outside of vision.